# Coating Strategy for Surface Modification of Stainless Steel Wire to Improve Interfacial Adhesion of Medical Interventional Catheters

**DOI:** 10.3390/polym12020381

**Published:** 2020-02-08

**Authors:** Zhaomin Li, Haijuan Kong, Muhuo Yu, Shu Zhu, Minglin Qin

**Affiliations:** 1State Key Laboratory for Modification of Chemical Fibers and Polymer Materials, College of Materials Science and Engineering, Donghua University, Shanghai 201620, China; zmli@accupathmed.com; 2Medical (Jiaxing) Co., Ltd., Jiaxing 314000, China; 3InnovaPath MedTech (Shanghai) Co., Ltd., Shanghai 201620, China; 4School of Materials Engineer, Shanghai University of Engineer Science, Shanghai 201620, China; Konghaijuan@sues.edu.cn

**Keywords:** stainless steel wire, surface modification, coating, braid-reinforced composite hollow fiber tube, interfacial adhesion

## Abstract

Poor interfacial bonding between stainless steel wire and the inner and outer layer resin matrix significantly affects the mechanical performance of braid-reinforced composite hollow fiber tube, especially torsion control. In this work, a coating of thermoplastic polyurethane (TPU) deposited on the surface of stainless steel wire greatly enhanced the mechanical performance of braid-reinforced composite hollow fiber tube. This method takes advantage of the hydrogen bonding between polyether block amide (PEBA) and thermoplastic polyurethane (TPU) for surface modification of stainless steel wire, as well as the good compatibility between PEBA and TPU. The mechanical properties of composited tubes demonstrate that the interlaminar shear strength, modulus of elasticity, and torque transmission properties were enhanced by 27.8%, 42.1%, and 41.4%, respectively. The results indicating that the interfacial adhesion between the coated stainless steel wire and the inner and outer matrix was improved. In addition, the interfacial properties of composite hollow fiber tube before and after coating was characterized by the optical microscope, and results show that the interfacial adhesion properties of the modified stainless steel wire reinforced resin matrix composites were greatly improved.

## 1. Introduction

With the development of minimally invasive interventional therapy, the application of medical interventional catheters has become more widespread. In the process of interventional treatment, it is necessary to deliver the catheter to the target site safely and effectively. Therefore, there are stringent requirements for the mechanical properties of the catheter, the most important of which are torque transmission and flexibility [1,2,3]. At present, the main method is to enhance the bursting strength and torque transmission of hollow fiber composite tubes by braiding reinforcement. Braid-reinforced composite hollow fiber tubes have excellent axial stiffness, radial support, and torsion control performance, which can reduce the operation time for the interventional catheter and the patient’s pain. According to the needs of different interventional treatments, different types of materials are selected, mainly materials reinforced with organic aramid fiber, inorganic glass fiber and carbon fiber, and metal wire [4,5,6].

Among these fiber-reinforced materials, the excellent ductility of stainless steel wire enables it to absorb more impact energy during the fiber crushing process and have higher deformability. At the same time, due to its excellent fatigue resistance and low creep, steel wire braided-reinforced tubes have better operating performance. Compared to carbon fiber and other reinforcing materials, stainless steel wire has slightly lower breaking strength and lower cost [7,8,9,10]. Based on these advantages, stainless steel wire has been widely applied in fiber-reinforced composites. Braided-reinforced composite tube essentially acts as a composite material, composed of inner and outer layers of a base material, and an intermediate braided layer, as shown in Figure 1a. However, the joints of stainless steel wire make it easy for slippage to occur between the wire and the inner and outer resin (Figure 1b), which leads to a reduction of the torque transmission performance. The shear modulus is the main factor that determines the torsional strength of hollow fiber tubes. The higher the shear modulus, the better the torsional strength. Currently, the torsional strength of composite hollow fiber tubes is mainly adjusted by the design of the braided structure, such as increasing the strength, number of strands, number of layers of braided stainless steel wire, and so on. However, as in other composite materials, the interfacial interaction is the key factor influencing the mechanical performance of the materials. The resin and fibers are bonded to each other via a two-phase interface, which acts as a bridge between the reinforcement and the matrix, playing a role in stress transmission [11,12].

Different methods of modifying the surfaces of fibers have been investigated as strategies to increase the interfacial strength between fiber [13], such as chemical etching [14,15] and grafting [16,17], plasma [18,19], addition of graphene [20], and so on. However, these methodologies often lead to a loss of tensile strength of the fibers and generate environmental pollution [21]. Coating is an effective approach to enhance the interface performance between the fiber and the resin matrix [21]. Zhu et al. focused on improving fiber–matrix interfacial interactions for carbon fiber (CF)/polyether ether ketone (PEEK) composites by introducing interfacial layers of polyether ketone ketone (PEKK) on activated CF, and the interlaminar shear strength, flexural strength, and modulus of CF/PEEK composites increased by 70%, 37%, and 48%, respectively [22]. Zhu et al. developed a novel strategy for surface modification of CF to improve interfacial interactions between CF and PEEK, and the interlaminar shear strength, flexural strength, and modulus of CF/PEEK composites increased by 71%, 63%, and 70%, respectively [23]. The coating and its application method are extremely important in interface optimization design [24,25,26]. Meanwhile, because it is difficult for high-performance thermoplastic resin to enter the fiber braid, it is necessary to modify the surface of the fiber. The main reasons are that stainless steel wire are chemically inert and hydrophobic with inadequate active groups, which lead to poor interfacial strength between fiber reinforcements and polymeric matrix. However, to the best of our knowledge, surface modification of stainless steel wire to improve the interfacial adhesion property of braid-reinforced composite tube to enhance interfacial adhesion of the fiber-reinforced tubes has not been reported. Based on this idea, a method of dip coating on stainless steel wire to improve the interfacial adhesion led to welding braided wires together at a high temperature, which are difficult to slide (Figure 1c).

In this paper, we propose a coating strategy for the modification of stainless steel wire to improve the interfacial adhesion of the fiber-reinforced tubes of medical interventional catheters. The stainless steel wire was coated with thermoplastic polyurethane (TPU) solution, then the weaving and coating extrusion process was started. The wall thickness of the coating of wire was systematically discussed. FTIR spectra were used to characterize the chemical structure of the surface after coating. The mechanical properties of the fiber-reinforced tube catheters were determined by interlaminar shear strength, modulus of elasticity, and torque transmission properties, and the results have indicated that they increased by 27.8%, 42.1%, and 41.4%, respectively. The results indicating that the interfacial adhesion between the coated stainless steel wire and the inner and outer matrix was improved. In addition, the interfacial properties of composite hollow fiber tube before and after coating was characterized by the optical microscope, and the results show that the interfacial adhesion properties of the modified stainless steel wire reinforced resin matrix composites were greatly improved. Due to TPU’s lower cost and better compatibility with the base material, it became our choice of coating modification material. This study also provides a new train of thought on improving the performance of medical interventional catheters in the future.

## 2. Materials and Methods

### 2.1. Materials

TPU (Pellethane TPU 2363-85A) was purchased from American Lubrizol Co., Ltd. (Cleveland, OH, USA). Masterbatch (Pantone 2727C±1C) was purchased from Clariant Chemicals Co., Ltd. (Holden, MA, USA). Stainless steel wire was supplied by American Fortwayn Co., Ltd. (Fort Wayne, IN, USA). Polyether block amide (PEBA) (6333) grades were supplied by Arkema (Philadelphia, PA, USA) in pellet form, with PA-12 as the hard segment and poly(tetramethylene oxide) (PTMO) as the soft segment. All resins were dried at 90 °C in an oven overnight before being used. The surface morphology of a cross-section of braid-reinforced composite hollow fiber tube was analyzed by 3D optical image (Accura C 320, Shenzhen Yuchuang Technology Co., Ltd, Shenzhen, China).

### 2.2. TPU for Surface Modification of Stainless Steel Wire

TPU solutions with different viscosities (250 mpa·s, 350 mpa·s) were poured into a dipping bath. The dried stainless steel wires were dipped vertically into the TPU solution and withdrawn at different rates (1.0, 1.5, 2.0, 2.5, 3.0, 3.5, 4.0, 4.5, 5.0 cm/min), then the coated wires were dried in a sintering oven. A schematic diagram of the dip coating process is shown in in Figure 2.

### 2.3. Preparation of Braid-Reinforced Composite Hollow Fiber Tube

The PEBA (6233) was dried in a circulating air oven at 90 °C for 24 h. In the extrusion process, a single screw with a diameter of 19 mm and an L:D ratio of 24:1 was used. The braid-reinforced composite hollow fiber tubes were prepared with different steel stainless wires (unmodified and modified).

### 2.4. Characterization

The surface functional groups of the stainless steel wire after coating with TPU was evaluated by using a Fourier transform infrared spectrometer (FTIR) (Nicolet 8700, Thermo Scientific, Waltham, MA, USA). The thickness of the coating was measured by laser caliper (DG2030, Proton, UK). The tensile test was estimated by an Instron machine (5943U3667, Instron, Boston, MA, USA) at room temperature with a crosshead speed of 625 mm/min. Interlaminar shear strength (ILSS) of composites was measured according to the ASTM D72644 system (5943U3667, Instron, Boston, MA, USA). Flexural properties of braid-reinforced composite hollow fiber tube were measured using an ASTM D2344 system (5943U3667, Instron, Boston, MA, USA). More than 5 parallel samples were used for each test, and results were averaged for final results. Standard deviation was indicated by error bars. The dynamic melting of the stainless steel wire after being coated was observed by a hot-stage polarizing microscope (BX51, Olympus, Japan). The torsion control performance of braid-reinforced composite hollow fiber tube was measured by homemade torsion control test equipment (Figure 3), the sample was passed through 2 parallel plates, and the angle of rotation of the distal end when rotated and 360° (20°/s) at the proximal end, which was connected to the proximal end of the motor torque to rotate counterclockwise 360° after 1 min was measured.

## 3. Results

### 3.1. Spectroscopy Analysis

Figure 4 shows the FTIR spectra of the stainless steel wire after being coated with TPU. The absorption peak at 3325 cm^−1^ was attributed to an N–H stretching vibration and hydrogen bonding stretching vibration. The absorption band at 1739 cm^−1^ was related to the stretching vibration of C=O groups, and the peak at 1542 cm^−1^ was ascribed to the bending vibration of N–H and stretching vibration of C–N. The bands at 2934 and 2854 cm^−1^ were ascribed to C–H stretching vibration. The absorption bands at 1244 cm^−1^ and 1043 cm^−1^ were ascribed to the symmetric and asymmetric vibrational stretching of C–O–C. These results indicate that TPU was successfully coated on the surface of the stainless steel wire.

### 3.2. Coating Thickness Calculations

The thickness of the coating has an important effect on the torsional properties of the braided wire. Therefore, it is very important to study different coating thicknesses to regulate the torsional properties of the composite tube. The influence factor of the thickness of coating is calculated as follows:h = 0.94 × (η × V)^2/3^/γ_lv_ 1/6(ρg)1/2(1)
where h is the thickness of the coating, ρ is the density, g is the gravity constant, V is the dragging speed, γ_lv_ is the vapor surface tension, and η is the viscosity. Table 1 shows the influences of different dragging speeds on the thickness of the coating. It can be seen from Table 1 that with increased dragging speed at the same viscosity, the coating thickness became larger. However, when the speed reached 4.0 m/min, the coating thickness reached the maximum. With an increased rate, the thickness of the coating had almost no change. This is because both the surface area of stainless steel wire and the solid content that can be accepted are certain. Meanwhile, it can be found from Table 1 that the coating thickness increased with viscosity at the same speed, which is consistent with Equation (1).

### 3.3. Interlaminar Shear Strength Analysis

Interlaminar shear strength tests were conducted to investigate the influence before and after coating on the interfacial adhesion properties of composite hollow fiber tube. As shown in Figure 5, with increased dragging speed, the interlaminar shear strength of composites also increased, indicating that the interfacial adhesion between the coated stainless steel wire and the inner and outer matrix was improved. The interfacial adhesion of composited tubes demonstrated that the interlaminar shear strength was enhanced by 27.8% (V = 5.0) after modification by TPU coating (V = 5.0). The possible mechanism, as depicted in Figure 6, is the intermolecular hydrogen between TPU and PEBA, resulting in improved interfacial adhesion between the resin matrix and the coated stainless steel wire. In addition, the coating has good compatibility with the resin matrix of the inner and outer layers. The resin is bonded by molten flow under high temperature and forms an interface layer after cooling, which is beneficial to improve the interfacial bonding strength of composites. Meanwhile, charge transfer interaction and π–π interaction between TPU and PEBA matrix led to good compatibility and wettability. In addition, mechanical interlocking was formed via penetration of TPU melt into the PEBA melt.

### 3.4. Interfacial Adhesion Analysis

In order to research the melting flow behavior of braid-reinforced stainless steel wire during extrusion, we simulated the melt flow behavior of coated modified stainless steel wire under heating conditions. Figure 7 shows the melting flow behavior of the coating on the surface of stainless steel wire during heating, observed by polarized optical microscopy in the heating stage. We designed two kinds of fiber arrangement: cross-lap (Figure 7a,b) and parallel (Figure 7c,d). Figure 7 shows the flow behavior of the coating on cross-lapped and parallel arrangements wires heated at 240 °C. It can be seen that as the coating gradually melted, the braided wires were fixed at the nodes and the sliding resistance between metal wires increased, which led to improved torque transmission capacity of the braided reinforced composite hollow fiber tube.

### 3.5. Torsion Control Performance

Torsion control performance is an important index to measure hollow fiber tubes for medical use. Figure 8 show the rotation angle of braid-reinforced composite hollow fiber tubes with coated and uncoated stainless steel wire under different dragging speeds. It can be seen in Figure 8 that the twist angle of the hollow fiber tube increased as the dragging speed increased. When the speed reached 4.5 cm/min, the angle of clock direction and the reverse rotation angle increased by 14.8% and 41.4%, respectively. This was due to the strong interfacial bonding strength between the coating-modified stainless steel wires and the inner and outer resin, resulting in increased torsional angle.

### 3.6. Tensile Properties of Braid-Reinforced Composite Hollow Fiber Tube

The results of mechanical measurements of braid-reinforced composite hollow fiber tubes are shown in Figure 9. It can be seen in Figure 9a,c that the modulus of elasticity and tensile strain increased by 42.1% and 31.5%, respectively. However, there was almost no change in tensile strength (Figure 9b). This demonstrates that the effect of the coating can improve the interfacial bonding between the stainless steel wire braid and the resin matrix. The improvement can be attributed to the intensive interactions between wire and matrix obtained from the hydrogen bond and good compatibility between TPU and PEBA. The mechanism might be attributed to the improved interfacial adhesion between stainless steel wire and resin matrix, as the result of a synergetic effect between TPU and PEBA matrix, resulting in stronger interface and enhanced mechanical performances.

### 3.7. Morphology of Braid-Reinforced Composite Hollow Fiber Tube

Figure 10 shows the tensile morphology of braid-reinforced composite hollow fiber tubes with coated and uncoated stainless steel wires measured by microscope. It can be seen from Figure 8b that the coated stainless steel wire did not undergo severe distortion. This is because the resin is tightly bonded to the wire to ensure the synchrony of tensile deformation. Uncoated wires were loose and bent after drawing. Therefore, the interfacial adhesion between the uncoated stainless steel wires and the resin matrix was poor. These results indicate that the interfacial adhesion of the stainless steel wire after coating with TPU and the resin matrix was greatly improved.

Figure 11 shows the surface morphology of cross-sections of braid-reinforced composite hollow fiber tubes with coated and uncoated stainless steel wires under the 3D optical microscopy. It can be seen in Figure 11a that uncoated wire and resin have a large gap between inner and outer tubes. The gap between the braided wire and the resin matrix causes the fiber tube to twist in the process of torsion, which leads to decreased torsion control performance. This indicates poor interfacial adhesion between the wire and the resin matrix. However, it can be seen from Figure 11b that the wire was closely bonded with the resin matrix. The indicates greatly improved interface adhesion between the coated stainless steel wire and the resin matrix.

### 3.8. Flexural Strength of Braid-Reinforced Composite Hollow Fiber Tube

Figure 12 shows the flexural strength of braid-reinforced composite hollow fiber tube before and after modification. It can be seen that there was almost no change in the flexural strength of the stainless steel wire, which ensures the flexibility of the composite tube and its ability to be used as a medical intervention tube, which is beneficial for doctors.

## 4. Conclusions

In this study, an efficient method for modifying interface adhesion between stainless steel wire and inner and outer layer resin matrix was established by introducing TPU as an interfacial layer surrounding the wire. The intermolecular hydrogen bonds formed between NH or C=O functional groups on the surface of TPU and NH or C=O functional groups in PEBA resulted in much stronger interactions between TPU layers and the resin matrix of the inner and outer layers. Meanwhile, TPU had good compatibility with the resin matrix of the inner and outer layers. The mechanical properties of fiber-reinforced medical interventional catheter tubes were measured by interlaminar shear strength, modulus of elasticity, and torque transmission properties. The results indicate greatly improved interfacial adhesion of the stainless steel wire and the resin matrix of the inner and outer layers. The interlaminar shear strength, modulus of elasticity, and torque transmission properties increased by 27.8%, 42.1%, and 41.4%, respectively, after modification by TPU coating. Our modification method provides a new train of thought on improving the performance of medical interventional catheters in the future. This approach can be utilized to improve interfacial interaction in high-temperature thermoplastic composites and has potential applications in high-performance polymer composites with excellent interfacial properties.

## Figures and Tables

**Figure 1 polymers-12-00381-f001:**
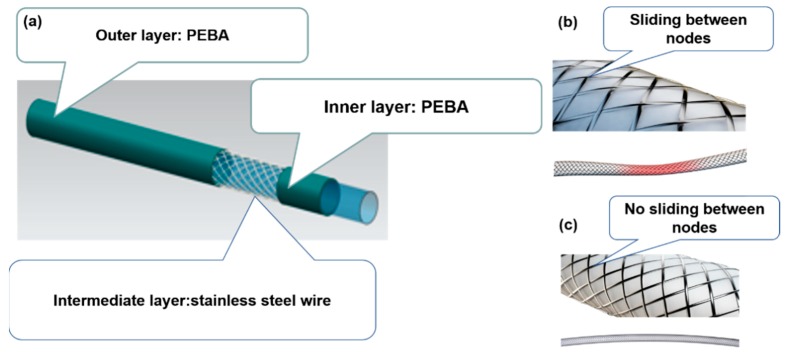
Braided reinforced composite tube. PEBA, polyether block amide, (**a**) composed of inner and outer layers of a base material and an intermediate braided layer, (**b**) sliding between nodes, (**c**) no sliding between nodes.

**Figure 2 polymers-12-00381-f002:**
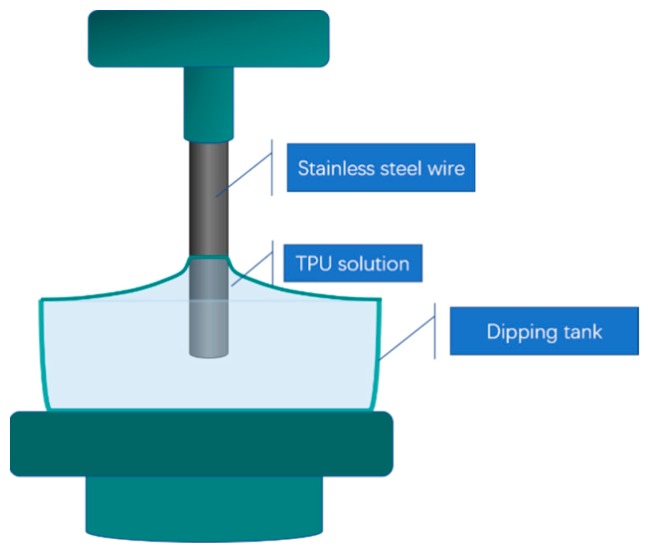
Schematic diagram of the dip- coating process.

**Figure 3 polymers-12-00381-f003:**
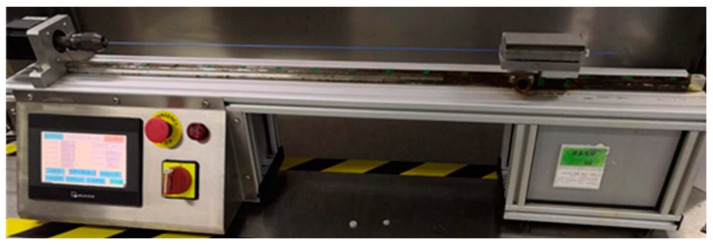
Torsion control test equipment.

**Figure 4 polymers-12-00381-f004:**
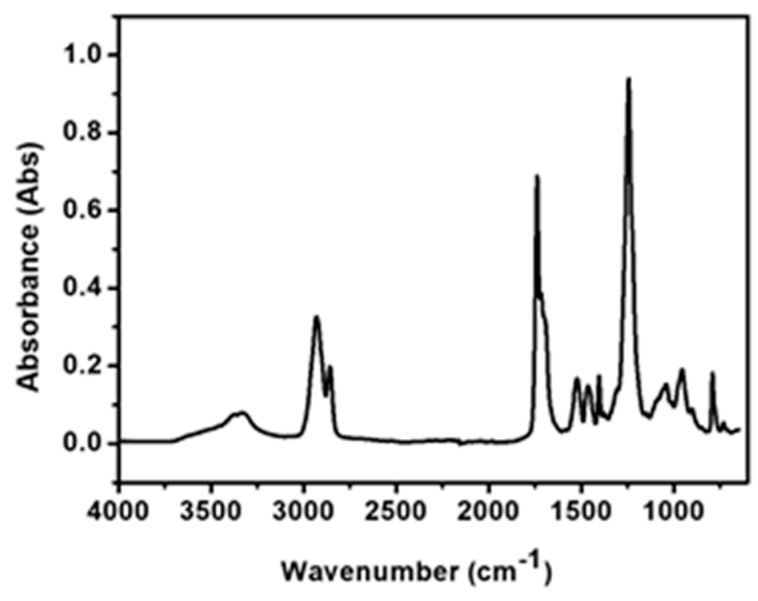
FTIR spectra of stainless steel wire after being coated with TPU.

**Figure 5 polymers-12-00381-f005:**
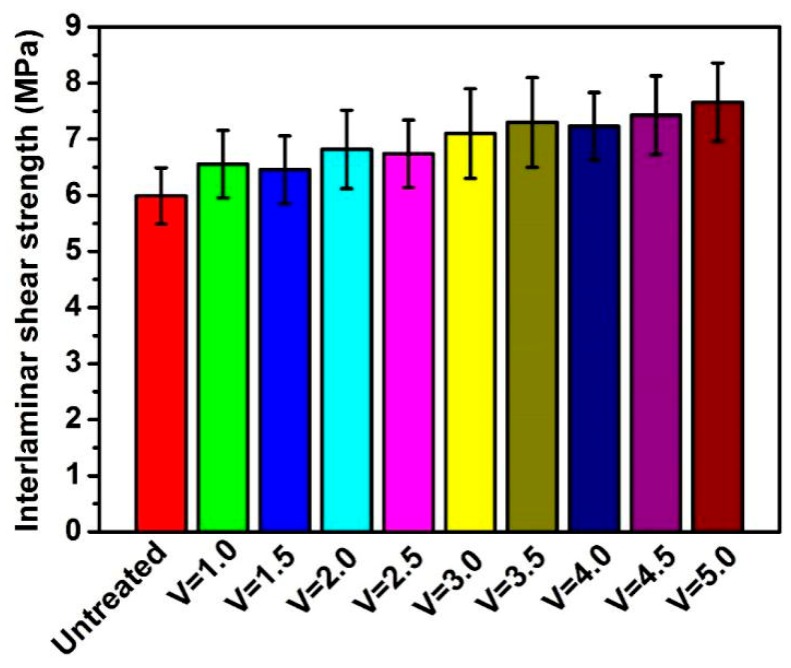
Interlaminar shear strength of composite before and after coating.

**Figure 6 polymers-12-00381-f006:**
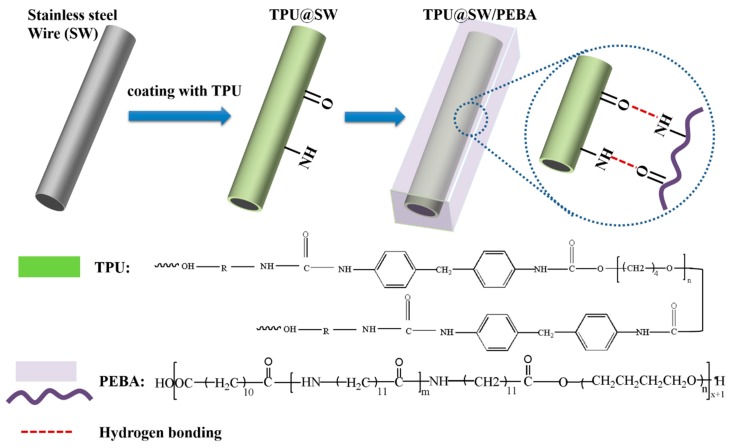
Possible mechanism of adhesion between PEBA and stainless steel wire by TPU coating.

**Figure 7 polymers-12-00381-f007:**
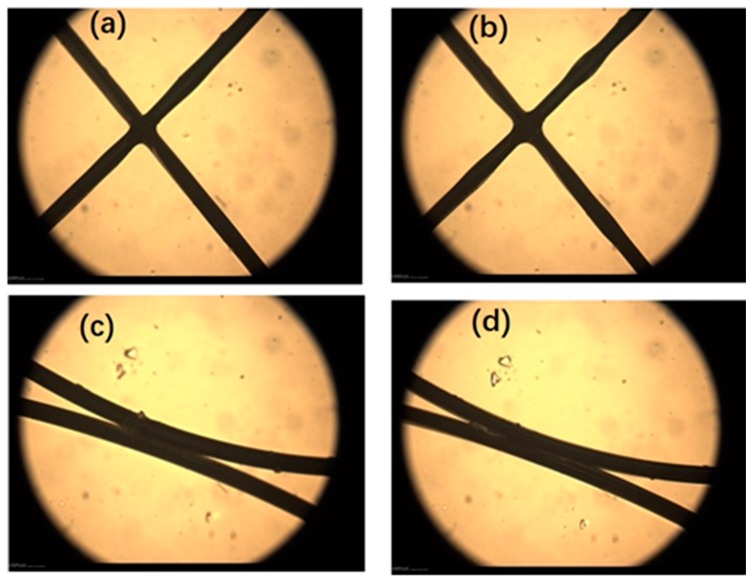
Interface adhesive process between coating of stainless steel wires by polarized optical microscopy, (**a**, **b**) melting flow behavior of the coating on the surface of stainless steel wire during heating in cross-lap arrangement, (**c**, **d**) melting flow behavior of the coating on the surface of stainless steel wire during heating in parallel arrangement.

**Figure 8 polymers-12-00381-f008:**
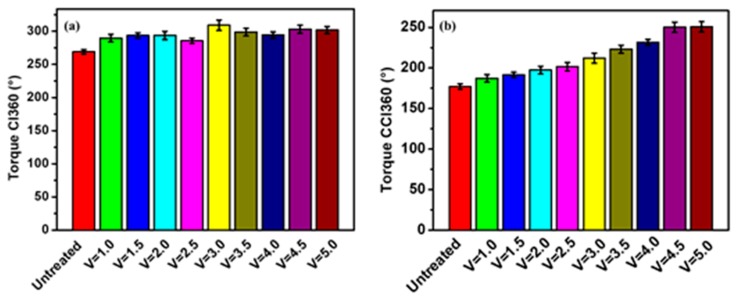
Rotation angle of braid-reinforced composite hollow fiber tube with coated and uncoated stainless steel wire: (**a**) clockwise angle, (**b**) counterclockwise angle.

**Figure 9 polymers-12-00381-f009:**
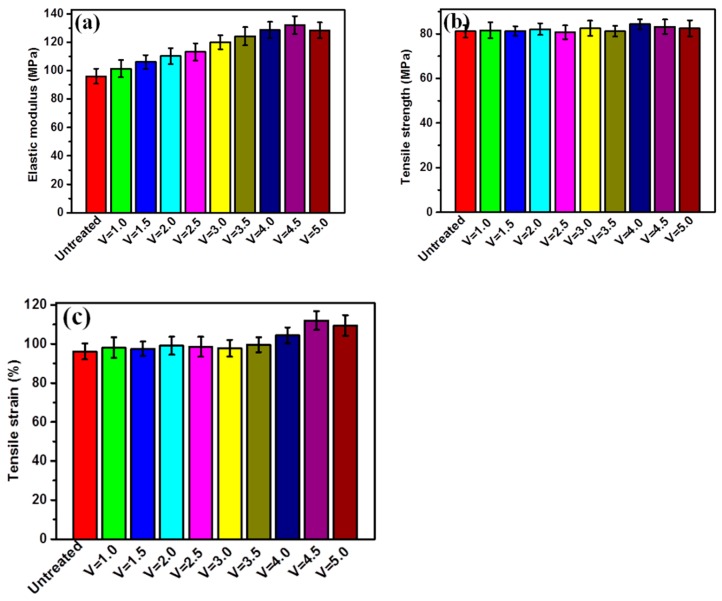
Tensile properties of braid-reinforced composite hollow fiber tubes with different coating speeds and uncoated stainless steel wires: (**a**) modulus of elasticity, (**b**) tensile strength, (**c**) tensile strain.

**Figure 10 polymers-12-00381-f010:**
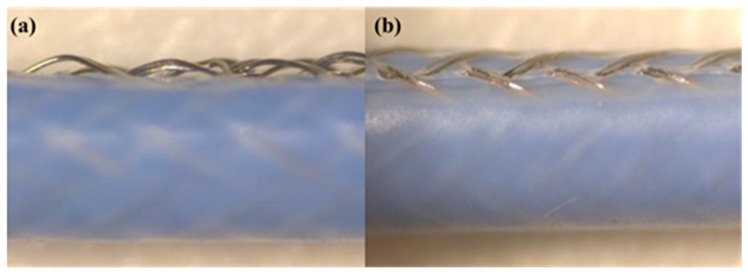
Surface morphology after tension of braid-reinforced composite hollow fiber tube with (**a**) uncoated and (**b**) coated stainless steel wires.

**Figure 11 polymers-12-00381-f011:**
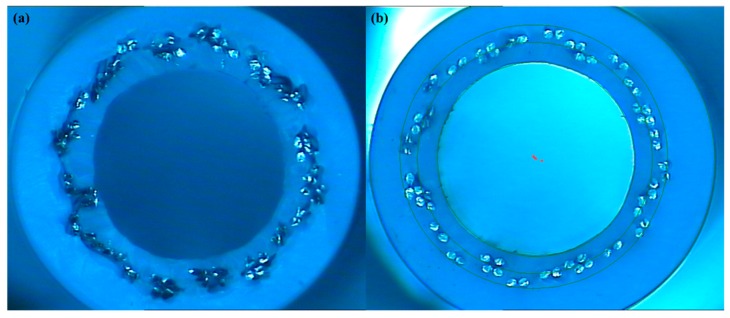
Surface morphology of cross-sections of braid-reinforced composite hollow fiber tubes with (**a**) uncoated and (**b**) coated stainless steel wires.

**Figure 12 polymers-12-00381-f012:**
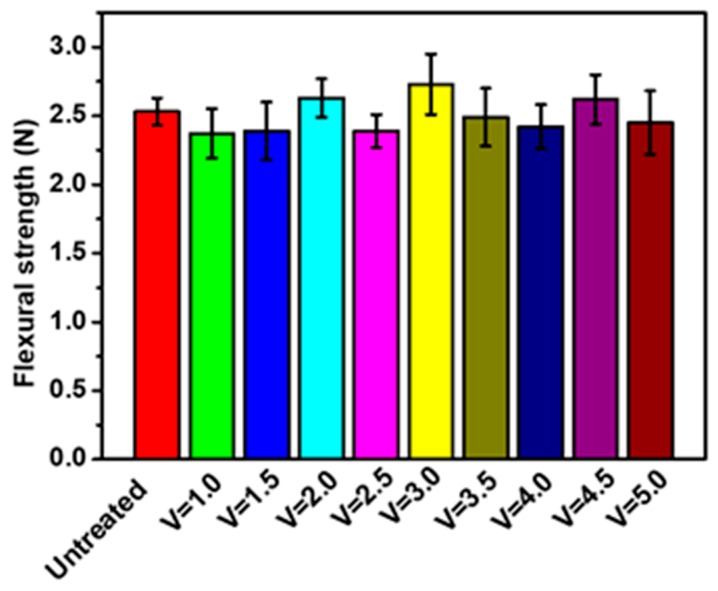
Flexural strength of braid-reinforced composite hollow fiber tubes with different coating speeds and uncoated stainless steel wires.

**Table 1 polymers-12-00381-t001:** Coating wall thickness with different speed and viscosity.

V1 (m/min), η1 = 250 mpa·s	h1 (μm)	V2 (m/min), η2 = 350 mpa·s	h2 (μm)
1.0	4.70	1.0	5.30
1.5	4.80	1.5	5.30
2.0	4.84	2.0	5.50
2.5	5.20	2.5	5.60
3.0	5.50	3.0	6.20
3.5	7.10	3.5	7.70
4.0	7.40	4.0	8.20
4.5	7.40	4.5	8.20
5.0	7.40	5.0	8.20

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
