# Peer review of "Coating Strategy for Surface Modification of Stainless Steel Wire to Improve Interfacial Adhesion of Medical Interventional Catheters"

_polymers, 2020, doi:10.3390/polym12020381_

Round 1

Reviewer 1 Report

General comments

The authors should comply with the journal's template. The spacing of the affiliations, references, etc. is not the appropriate one. In addition, the paper would benefit from some closer proofreading.

Specific comments

Abstract: All acronyms should be defined upon first appearance in the text. For instance, polyether block amide (PEBA) and thermoplastic polyurethane (TPU) should be clarified.

L58. Figure 1 caption. Explain what (a), (b) and (c) are.

L66. carbon fiber (CF)

L80. thermoplastic polyurethane (TPU)

L81. Fourier-Transform Infrared Spectroscopy (FTIR)

L84-85. It makes no sense to include results in the introduction.

L92. Polyether Block Amide (PEBA)

L103. Delete Figure 2 (or move it to supporting information).

L125. Figure 3 should be moved to supporting information or should be deleted.

L158. Pebax > PEBA (be consistent in the use of acronyms and avoid trademarks)

Figure 7. Explain what it (a), (b), (c) and (d). I cannot see the difference between (a) and (b), and between (c) and (d). Please explain.

L199. Tesile > Tensile

L202. Reinforcd > reinforced.

Author Response

Title changed to:Coating Strategy for Surface Modification of Stainless Steel Wire to Improve Interfacial Adhesion of Medical Interventional Catheters

L58. Figure 1 caption was explain to:(a) composed of inner and outer layers of a base material and an intermediate braided layer, (b) sliding between nodes, (c) no sliding between nodes

L66. CF was explain to carbon fiber (CF)

L81 was modify to Fourier-Transform Infrared Spectroscopy (FTIR)

L84-85 was modify to the mechanical properties of composited tubes demonstrate that the interlaminar shear strength, modulus of elasticity, and torque transmission properties were enhanced by 27.8%, 42.1%, and 41.4%, respectively.

L92 was expiain to Polyether Block Amide (PEBA)

L103.The Figure 2 was deleted

L125. The Figure 3 was deleted

L158 was modify to PEBA

Figure 7 see tha article

L199. Tesile was modify to tensile

L202. Reinforcd was modify to reinforced.

Reviewer 2 Report

The manuscript entitled “Coating of Strategy for Surface Modification Stainless Steel Wire to Improve Interfacial Adhesion of Medical Interventional Catheters” by Li and coworkers describes the deposition of a thermoplastic polyurethane (TPU) onto stainless steel wires via a dip-coating process.  This method took advantage of hydrogen bonding interactions between PEBA and the TPU as well as the good compatibility between these two materials.

            Overall, this work is well characterized and the findings are generally well-supported by the experiments performed by the authors.  A variety of characterizations were performed, including infrared spectroscopy as well as optical microscopy observations,  in addition to mechanical property tests such as shear strength, flexural strength, interfacial adhesion analysis, and torsion control tests.  Possibly scanning electron microscopy (SEM) characterization could also be of interest, but the characterizations appear to be sufficient as presented by the authors.  

            The work presented is highly relevant both from a fundamental scientific perspective and from an applied perspective, and this material would be of interest to readers of Polymers.  Although I believe the scientific work conducted by the authors is of a high quality, the writing of this manuscript is somewhat unclear and it seems to be difficult to follow in its current form.  Significant proofreading or polishing, is therefore required.    

Some examples (but not all) of where some polishing is needed are provided below: 

Lines 2-4: The wording or writing of the title of this manuscript is unclear and needs revision or polishing.

Lines 16-17: The sentence “Due to poor interfacial bonding between stainless steel wire and inner and outer layer resin matrix, significantly affecting mechanical performances of braid reinforced composite hollow fiber tube, especially the torsion control performance” is somewhat difficult to follow and unclear.

Line 18: The abbreviation TPU is mentioned in the abstract but a definition of this abbreviation does not seem to be provided.

Line 20: “hydrogen bond” can be changed to “hydrogen bonding”.

Line 21: The abbreviation PEBA is mentioned in the abstract but a definition of this abbreviation does not seem to be provided.

Lines 30-31: “which are most important being torque transmission and flexibility” can possibly be changed to “of which the most important includes torque transmission and flexibility”.

Line 32: “of hollow fiber composite tube” can be changed to “of a hollow fiber composite tube”.

Line 39: “In these” can be changed to “Among these”.

Line 43: “On the basic of these” can be changed to “On the basis of these”.

Lines 47-48: The phrase “the joint of stainless steel wire are easy to occur to slip between the stainless steel wire and the inner and outer resin” is unclear.

Line 48: “which lead to reduce the torque transmission ability.” Can possibly be changed to “which leads to a reduction of the torque transmission performance.”

Lines 59-60: The phrase “Different methods have been investigated that the modification of the fiber surface is beneficial to increase the interfacial strength between the fiber“ is unclear.  Maybe this can be changed to “Different methods of modifying the surfaces of fibers have been investigated as strategies to increase the interfacial strength between the fiber”.

Line 62: “often lead to sacrificed the tensile strength of the fibers and environmental” can possibly be changed to “often lead reductions in the tensile strengths of the fibers and generate environmental”.

Lines 67-68: “for surface modification CF to improve” can be changed to “for the surface modification of CF to improve”.

Lines 71-72: The phrase “due to the high-performance thermoplastic resin is

difficult to enter the fiber braid,” is unclear.

Lines 73-74: The phrase “it has not been reported that surface modification of steel stainless wire to improve the interfacial adhesion property of braid reinforced composite tube to enhance torsional strength” is very important as it provides an emphasis on the significance of this research, but this phrase is unclear.

Lines 75-77: This sentence at the end of this paragraph is somewhat unclear.

Line 78: The phrase “we proposed coating of strategy for modification of stainless steel wire” can possibly be changed to “we proposed a coating strategy for the modification of stainless steel wire”.

Line 80: The phrase “then starting the weaving and coating extrusion process” is unclear.

Line 81: “spectra was used” can be changed to “spectra were used” or “spectroscopy was used”.

Lines 81-82: “for characterization the chemical structure of the surface” can possibly be changed to “to characterize the chemical structure of the surface”.

Line 98: The phrase “Different viscosity (250 mpa·s, 350 mpa·s) of the TPU solution was poured into dipping bath” can possibly be changed to “TPU solutions with different viscosities (250 mpa·s, 350 mpa·s) were poured into a dipping bath”.

Line 101: The phrase “Schematic diagram of the dip coating process are shown in “ can be changed to “A schematic diagram of the dip coating process is shown in”.

Line 116: “using ASTM D2344 system” can be changed to “using an ASTM D2344 system”.

Line 118: “stainless steel wire after coated were observed” can be changed to “stainless steel wire samples after they had been coated were observed” or “stainless steel wire samples after the coating treatment were observed”.

Lines 122-123: The phrase “which is connected to the proximal end the motor torque to rotate counterclockwise” seems to be a little unclear.

Line 129: “to N-H stretching vibration and hydrogen bonding” can be changed to “to an N-H stretching vibration and a hydrogen bonding”.

Line 131: Two sentences seem to be merged incorrectly in the phrase “groups And the peaks at 1542 cm-1 was”.  Possibly this can be changed to “groups. Meanwhile, the peak at 1542 cm-1 was.”

Line 135: “steel stainless” should be “stainless steel”.

Lines 143-144: “Table 1 shows the different dragging speed of the thickness of coating.” Can possibly be changed to “Table 1 shows the influences of different dragging speeds on the thickness of the coating.”.

Lines 147-148: The sentence “This is account of the surface area of stainless steel wire is certain and the solid content that can be accepted is certain” is unclear.

Line 149: “of coating” can be changed to “of the coating”.

Lines 151-152, Table 1:  Error margins or error ranges may be needed for the numerical values provided in Table 1.

Lines 157-159: The sentence “The possible mechanism asdepicted in Figure 6, the intermolecular hydrogen between TPU and Pebax, and resulting in improved interfacial adhesion between resin matrix and coating steel stainless wire” is unclear.”

Line 167: The phrase “As showed in Figure 7, the melting flow behavior” can possibly be changed to “Figure 7 shows the melting flow behavior”.

Line 169: “arrangement” can be changed to “arrangements”.

Line 171: “wire were heated at 240 °C, it can be found that with” can be changed to “wire were heated at 240 °C. It can be found that as”.

Lines 176-177, Figure 7 caption: Although descriptions for images (a)-(e) are provided in text prior to Figure 7.  Descriptions/definitions of the images (a)-(e) should be provided in the caption.

Line 192: “However, the tensile strength almost no change” can possibly be changed to “However, the tensile strength did not change significantly”.

Line 195: “obtained from hydrogen bond between” can possibly be changed to “obtained from hydrogen bonds between” or maybe “obtained via hydrogen bonding between”.

Line 204: “didn’t” can be changed to “did not”.

Lines 213-214: The sentence “It can be found from Figure 9a that uncoated wire and resin with a large gap.” is unclear.

Lines 215-216: “which lead to torsion control performance decrease.” Can possibly be changed to “which lead to a decrease in the torsion control performance.”.

Lines 220-221, Figure 11: The authors may need to indicate which type of microscopy or technique was used to obtain these images (for example, optical microscopy, SEM, or TEM?).  Scale bars would also be helpful.

Line 225: “was almost no change,” can possibly be changed to “was virtually unchanged,” or “did not change significantly,”.

Lines 225-227: The phrase “indicates that ensure the flexibility of the composite tube and transportability as a medical intervention tube, it is beneficial to the doctor's operation” is unclear.

Lines 233-234: “as interfacial layer surrounding stainless steel wire” can be changed to “as an interfacial layer surrounding a stainless steel wire”.

Line 234: “The intermolecular hydrogen bond formed between” can be changed to “Intermolecular hydrogen bonds formed between”.

Line 235: “group on surface of TPU and NH or C=O functional group” can be changed to “groups on the surface of TPU and NH or C=O functional groups”.

Line 242: “27.8%, 42.1% and 41.4%” The authors may need to provide error margins or ranges for these values. 

Author Response

Lines 2-4: The  title of this manuscript was modify to Coating Strategy for Surface Modification of Stainless Steel Wire to Improve Interfacial Adhesion of Medical Interventional Catheters

Lines 16-17 was changed to Poor interfacial bonding between stainless steel wire and the inner and outer layer resin matrix significantly affects the mechanical performance of braid-reinforced composite hollow fiber tube, especially torsion control

Line 18 was changed to  thermoplastic polyurethane (TPU)

Line 20: “hydrogen bond”was changed to “hydrogen bonding”

Line 21 was changed to polyether block amide (PEBA)

Lines 30-31: which are most important being torque transmission and flexibility was changed to the most important of which are torque transmission and flexibility

Line 32: “of hollow fiber composite tube” was changed to “of hollow fiber composite tubes

Line 39: “In these”was changed to “Among these”

Line 43: “On the basic of these” was changed to “ Based on these ”

Lines 47-48 was changed to  the joints of stainless steel wire make it easy for slippage to occur between the wire and the inner and outer resin (Figure 1b), which leads to a reduction of the torque transmission performance

Line 48: “which lead to reduce the torque transmission ability.” was changed to “which leads to a reduction of the torque transmission performance”

Lines 59-60: The phrase “Different methods have been investigated that the modification of the fiber surface is beneficial to increase the interfacial strength between the fiber“ was changed to “Different methods of modifying the surfaces of fibers have been investigated as strategies to increase the interfacial strength between the fiber”

Line 62: “often lead to sacrificed the tensile strength of the fibers and environmental” was changed to “ a loss of tensile strength of the fibers and generate environmental pollution”

Lines 67-68: “for surface modification CF to improve”was changed to “for the surface modification of CF to improve”

Lines 71-72 was changed to  because it is difficult for high-performance thermoplastic resin to enter the fiber braid

Lines 73-74 was changed to  surface modification of stainless steel wire to improve the interfacial adhesion property of braid-reinforced composite tube to enhance torsional strength has not been reported.

Lines 75-77 was changed to Based on this idea, a method of dip coating on stainless steel wire to improve the interfacial adhesion led to welding braided wires together at high temperature, which are difficult to slide

Line 78: The phrase “we proposed coating of strategy for modification of stainless steel wire” was changed to “we proposed a coating strategy for the modification of stainless steel wire”

Line 80 was changed to The stainless steel wire was coated with thermoplastic polyurethane (TPU) solution, then the weaving and coating extrusion process was started.

Line 81: “spectra was used” was changed to spectra were used

Lines 81-82: “for characterization the chemical structure of the surface” was changed to “to characterize the chemical structure of the surface”

Line 98: The phrase “Different viscosity (250 mpa·s, 350 mpa·s) of the TPU solution was poured into dipping bath”was changed to “TPU solutions with different viscosities (250 mpa·s, 350 mpa·s) were poured into a dipping bath”

Line 101: The phrase “Schematic diagram of the dip coating process are shown in was deleted

Line 116: “using ASTM D2344 system” was changed to “using an ASTM D2344 system”

Line 118: “stainless steel wire after coated were observed” was changed to  stainless steel wire after being coated was observed

Lines 122-123 was changed to the angle of rotation of the distal end when rotated and 360° (20°/sec) at the proximal end, which was connected to the proximal end of the motor torque to rotate counterclockwise 360° after 1 min, was measured

Line 129: “to N-H stretching vibration and hydrogen bonding” was changed to “to an N-H stretching vibration and a hydrogen bonding”

Line 131: Two sentences seem to be merged incorrectly in the phrase “groups And the peaks at 1542 cm-1 was”. was changed to groups, and the peak at 1542 cm–1 was ascribed to the bending vibration of N–H and stretching vibration of C–N

Line 135: “steel stainless” was changed to “stainless steel”.

Lines 143-144: “Table 1 shows the different dragging speed of the thickness of coating.” was changed to “Table 1 shows the influences of different dragging speeds on the thickness of the coating.”

Lines 147-148 was changed to . This is because both the surface area of stainless steel wire and the solid content that can be accepted are certain

Line 149: “of coating” was changed to the coating thickness increased with viscosity at the same speed

Lines 151-152:because the unit of wall thickness is micron, it is difficult to confirm the error

Lines 157-159 was changed to The possible mechanism, as depicted in Figure 4, is the intermolecular hydrogen between TPU and PEBA, resulting in improved interfacial adhesion between the resin matrix and the coated stainless steel wire

Line 167: The phrase “As showed in Figure 7, the melting flow behavior”was changed to “Figure 7 shows the melting flow behavior”

Line 169: “arrangement” was changed to “arrangements”

Line 171: “wire were heated at 240 °C, it can be found that with” was changed to “wire were heated at 240 °C. It can be found that as”

Lines 176-177 see the article

Line 192: “However, the tensile strength almost no change”was changed to “However, there was almost no change in tensile strength

Line 195 see the article

Line 204: “didn’t”was changed to “did not”

Lines 213-214 was changed to t can be seen in Figure 9a that uncoated wire and resin have a large gap between inner and outer tubes.

Lines 215-216: “which lead to torsion control performance decrease.”was changed to which leads to decreased torsion control performance

Lines 220-221 see the article

Line 225: “was almost no change,” was changed to  was almost no change in the flexural strength of the stainless steel wire

Lines 225-227 see the article

Lines 233-234: “as interfacial layer surrounding stainless steel wire”was changed to as an interfacial layer surrounding the wire

Line 234: “The intermolecular hydrogen bond formed between”was changed to “Intermolecular hydrogen bonds formed between”

Line 235: “group on surface of TPU and NH or C=O functional group” was changed to “groups on the surface of TPU and NH or C=O functional groups”

Line 242: Figure 7 provides the  error margins

Round 2

Reviewer 2 Report

  Overall, I believe that the authors of “Coating Strategy for Surface Modification of Stainless Steel Wire to Improve Interfacial Adhesion of Medical Interventional Catheters” (Manuscript ID: polymers-685026) have successfully addressed the recommendations of the reviewers.  In my opinion, the manuscript is greatly improved and suitable for publication.

I have included a few minor comments or possible suggestions below.   

Line 23: the font of the text for the word “mechanical” is not consistent with the other text in this manuscript.

Line 109: “TPU solution” can possibly be changed to “TPU solutions”.

Line 115: The caption for the Figure or diagram on page 4 seems to be missing.

Line 138: The caption for the Figure or diagram on page 4 seems to be missing.

Author Response

Line 23: the font of the text for the word “mechanical” is not consistent with the other text in this manuscript: see the article.

Line 109: TPU solution: It was changed to TPU solutions.

Line 115: The caption for the Figure or diagram on page 4 seems to be missing: I have re-added it.

Line 138: The caption for the Figure or diagram on page 4 seems to be missing: I have re-added it.